# Lipoprotein(a)—The Crossroads of Atherosclerosis, Atherothrombosis and Inflammation

**DOI:** 10.3390/biom12010026

**Published:** 2021-12-24

**Authors:** Sabina Ugovšek, Miran Šebeštjen

**Affiliations:** 1Faculty of Medicine, University of Ljubljana, 1000 Ljubljana, Slovenia; ugovsek.sabina@gmail.com; 2Department of Cardiology, University Medical Centre Ljubljana, 1000 Ljubljana, Slovenia; 3Department of Vascular Diseases, University Medical Centre Ljubljana, 1000 Ljubljana, Slovenia

**Keywords:** lipoprotein(a), atherothrombosis, inflammation, coagulation, fibrinolysis

## Abstract

Increased lipoprotein(a) (Lp(a)) levels are an independent predictor of coronary artery disease (CAD), degenerative aortic stenosis (DAS), and heart failure independent of CAD and DAS. Lp(a) levels are genetically determinated in an autosomal dominant mode, with great intra- and inter-ethnic diversity. Most variations in Lp(a) levels arise from genetic variations of the gene that encodes the apolipoprotein(a) component of Lp(a), the *LPA* gene. *LPA* is located on the long arm of chromosome 6, within region 6q2.6–2.7. Lp(a) levels increase cardiovascular risk through several unrelated mechanisms. Lp(a) quantitatively carries all of the atherogenic risk of low-density lipoprotein cholesterol, although it is even more prone to oxidation and penetration through endothelia to promote the production of foam cells. The thrombogenic properties of Lp(a) result from the homology between apolipoprotein(a) and plasminogen, which compete for the same binding sites on endothelial cells to inhibit fibrinolysis and promote intravascular thrombosis. *LPA* has up to 70% homology with the human plasminogen gene. Oxidized phospholipids promote differentiation of pro-inflammatory macrophages that secrete pro-inflammatory cytokines (e. g., interleukin (IL)-1β, IL-6, IL-8, tumor necrosis factor-α). The aim of this review is to define which of these mechanisms of Lp(a) is predominant in different groups of patients.

## 1. Introduction

Lipoprotein(a) (Lp(a)) is a complex plasma protein that consist of low-density lipoprotein (LDL) cholesterol and apolipoprotein B-100 (apoB) linked to the plasminogen-like apolipoprotein(a) (apo(a)) via a disulfide bond. Lp(a) levels are genetically determined by the *LPA* gene and they vary among individuals from ≤0.2 to ≥250 mg/dL, although within a single individual, Lp(a) levels are stable throughout life [1,2]. The *LPA* gene is located on chromosome 6q26–q27 and it encodes the highly glycosylated hydrophilic apo(a) [3,4]. Apo(a) shows high amino-acid sequence homology to serine protease plasminogen [3]. Apo(a) contains a protease-like domain and two tri-loop structures known as ‘kringles’ (KIV, KV) [1,3], and the KIV domain has 10 types [3]. The different numbers of KIV type 2 (KIV_2_) repeat in apo(a) results in Lp(a) isoform size heterogeneity, and more KIV_2_ repeats results in a larger apo(a) isoform. The larger apo(a) isoforms are less efficiently secreted from hepatocytes, and consequently there is an inverse correlation between the apo(a) isoform size and the Lp(a) levels in plasma [1,5]. Patients with a smaller apo(a) isoform size not only have higher Lp(a) levels, but also have significantly greater risk of coronary artery disease (CAD) [6]. The number of KIV_2_ repeats accounts for 69% of the variation in Lp(a) levels [7]. As well as the different number of KIV_2_ repeats, the Lp(a) isoform size and levels are determined by more than 200 single nucleotide polymorphisms (SNPs) in the wider *LPA* region. Among these SNPs, rs10455872 and rs3798220 have the most influence on Lp(a) levels [4], as they explain 36% of the variation in Lp(a) levels [8]. *LPA* variants are carried by one in six people, who have a 1.5-fold greater risk of coronary diseases [8]. rs10455872 and rs3798220 polymorphisms are also independent predictors of cardiovascular events in patients with CAD [8].

Increased Lp(a) levels are also a risk factor for degenerative aortic valve stenosis [9,10]. Both homozygote and heterozygote carriers of rs10455872 have increased risk of degenerative aortic valve stenosis, although this risk is not seen for carriers of rs3798220. A trend towards increased risk has been connected with lower numbers of KIV_2_ repeats [10].

Elevated levels of Lp(a) can occur in patients with otherwise normal lipid levels [4]. Dietary and environmental factors have minimal contributions to Lp(a) plasma levels [1]. A quarter of the general population has Lp(a) plasma levels >20 mg/dL, which have been associated with a 2-fold increased risk of developing cardiovascular disease. Although there is no conclusive evidence for specific Lp(a) cut-off points based on age, sex, and race, the European guidelines consider those with ≥50 mg/dL Lp(a) to be at high risk [11]. Lp(a) levels >180 mg/dL are correlated with risk of cardiovascular events similar to that of familial hypercholesterolemia [1].

According to genetic and epidemiological studies, Lp(a) is considered pro-atherosclerotic (Figure 1), pro-inflammatory (Figure 2), pro-thrombotic, and anti-fibrinolytic (Figure 3) [12].

## 2. Lipoprotein(a) and Atherosclerosis

Lipoprotein(a) can be internalized and accumulated in the intima of arteries and the aortic valve leaflets. Lp(a) enters the intima at similar rates to LDL cholesterol, although this does not occur via the lipoprotein receptor for LDL cholesterol, but is dependent on Lp(a) plasma concentrations, Lp(a) particle size, blood pressure, and arterial wall permeability [13,14]. Lp(a) accumulates all over the intima, whereas LDL cholesterol and other apoB-containing lipoproteins remain mainly at atherosclerotic lesions. Compared to LDLs, Lp(a) has a greater affinity for the vascular wall and for proteoglycans and fibronectin on the endothelial cell surface [13]. Furthermore, Lp(a) is more atherogenic than LDL cholesterol, because it consists of all of the atherogenic components of both LDL cholesterol and apo(a) [4,15,16].

Various studies have indicated that Lp(a) is taken up by macrophages to produce foam cells, and thus to promote the development of atherosclerotic lesions [13]. Atherosclerosis is an inflammatory disease of the arterial wall, whereby several events result in the formation of a complex atherosclerotic plaque that is composed of a lipid-rich core covered with a fibrous cap (Figure 1) [17]. Lp(a) promotes atherosclerotic plaque formation through various mechanisms. Lp(a) induces expression of inflammatory cytokines, including interleukin (IL)-1β, IL-6, IL-8, and tumor necrosis factor-α, and increases expression of adhesion molecules on the surface of the endothelial cells, including vascular cell adhesion protein-1, intercellular adhesion molecule-1, E-selectin, P-selectin, and others [3,18,19]. Lp(a) promotes monocyte chemotaxis via stimulation of secretion of monocyte chemotactic protein and activation of nuclear factor κB [20]. Additionally, Lp(a) binds and transports more than 70% of circulating oxidized phospholipids, which are involved in plaque vulnerability and destabilization [21].

As indicated, atherosclerotic plaques in patients with elevated Lp(a) levels have a complex morphology [17]. These plaques are prone to repeated ruptures and healing, which can lead to severe and rapid progression of atherosclerosis. Coronary atherosclerosis in patients with high Lp(a) levels mostly manifests clinically as an acute myocardial infarction, rather than stable angina [4]. Several large observational studies and meta-analyses have also reported gradual increased risk of CAD and ischemic stroke with increasing Lp(a) levels, without any threshold value defined [22,23,24,25]. This risk is higher in younger populations than older populations [25]. Indeed, the addition of Lp(a) levels to traditional risk score charts improve the accuracy of the prediction of susceptibility for future cardiovascular events. This has been shown by several studies of primary and secondary prevention [26,27,28], except for primary prevention in women with low risk of cardiovasculer events [29].

Statins are well known to reduce LDL cholesterol levels, but as seen in a meta-analysis of 29,069 patients on statin therapy, they do not provide any significant changes in Lp(a) levels [30]. Moreover, Lp(a) levels in patients treated with statins are more strongly associated with cardiovascular disease risk than for those on a placebo. The main explanation for this was that when LDL cholesterol levels are reduced, Lp(a)-attributable risk becomes an even stronger predictor of residual risk [30]. This observation has been supported by two interventional studies: ‘Further Cardiovascular Outcomes Research with PCSK9 Inhibition in Subjects with Elevated Risk’ (FOURIER) and ‘Evaluation of Cardiovascular Outcomes After an Acute Coronary Syndrome During Treatment with Alirocumab’ (ODYSSEY Outcomes) [31,32]. Patients on LDL cholesterol-lowering therapies and with high Lp(a) levels were associated with increased risk for cardiovascular disease independent of LDL cholesterol levels. Evolocumab therapy in the first of these two trials, and alirocumab therapy in the second, reduced Lp(a) levels and cardiovascular disease risk independent of LDL cholesterol lowering [31,32].

Patients with elevated Lp(a) have significantly more recurrent acute myocardial infarctions. One of the reasons for this is that Lp(a) specifically accumulates at injured sites, which results in restenosis after coronary artery revascularisation procedures [4]. A meta-analysis of 1834 patients reported a positive correlation between high Lp(a) levels and in-stent restenosis, especially in patients with drug-eluting stents [33]. Lowering Lp(a) levels with lipoprotein apheresis reduces restenosis rates and helps to prevent recurrent coronary syndromes [34,35].

As well as being a risk factor for coronary atherosclerosis, elevated Lp(a) levels define the risk of ischemic stroke, peripheral arterial disease, and calcific aortic valve disease. Calcific aortic valve disease is a multifactor condition and a dynamic process similar to atherosclerosis. Endothelial cell dysfunction followed by subendothelial accumulation of lipoproteins and chronic inflammation results in a condition that can vary from mild valve thickening to severe calcification with stenotic occlusion [36]. Studies have shown that Lp(a) levels ≥ 30 mg/dL represent a risk factor for calcific aortic valve disease. Lp(a) contributes to this disease at different rates, which depend on ethnicity [9], with the highest levels of Lp(a) in calcific aortic valve disease seen for Caucasians [9,37,38]. There is also an association between Lp(a) gene polymorphisms and calcific aortic valve disease [36,38,39,40]. Thanassoulis et al. and Ozkan et al. showed strong positive correlations for rs10455872 and rs3798220 polymorphisms with the progression of calcific aortic valve disease [36,40].

Patients with heterozygous familial hyperholesterolemia have significantly higher Lp(a) levels and hazard ratio for acute myocardial infarction, compared to those without, as was shown in the prospective population study, ‘Copenhagen General Population Study,’ and in a cross-sectional analysis of ‘Spanish Familial Hypercholesterolemia Cohort Study’ (SAFEHEART) [41,42]. Furthermore, patients with Lp(a) levels ≥50 mg/dL and LDL-receptor-negative mutations have a higher risk of cardiovascular disease, compared to patients with less severe mutations [42].

## 3. Lipoprotein(a) and Inflammation

Inflammation has a vital role in the development and progression of atherosclerosis, and thus it contributes to increased risk of cardiovascular events [43]. Inflammation promotes endothelial cell activation, dysfunction and loss of endothelial integrity, failure of endothelial repair, intima lipid deposition, and plaque formation and instability [44].

Despite the evidence that Lp(a) levels are genetically determined, several studies have shown that chronic inflammation interferes with Lp(a) expression and increases Lp(a) plasma levels [21]. On the other hand, as for other lipoproteins, Lp(a) is susceptible to oxidative modifications and formation of pro-inflammatory and pro-atherogenic oxidized phospholipids (OxPLs) [3]. Lp(a) carries more than 80% OxPLs in its particles, and consequently this increases the inflammatory activity of the arterial wall [21]. Taken together, the evidence shows a bidirectional link between Lp(a) and inflammation (Figure 2) [44].

The Lp(a) binding of OxPLs is to lysine residues on isolated fragments of KV of apo(a) [45]. At low plasma concentrations, Lp(a) has anti-inflammatory effects via its scavenging of OxPLs and their degradation by Lp(a)-associated phospholipase A_2_ (Lp-PLA_2_) [21]. At high Lp(a) levels, its anti-atherogenic effects are diminished, with decreased Lp-PLA_2_ lowering of oxidative stress, compared to LDL-associated PLA_2_. The OxPL moieties on apo(a) compete for the active site and decrease the catalytic efficiency of Lp-PLA_2_ [46]. Here, the oxidation–reduction state tends towards oxidation, and Lp(a) releases its oxidative load into the atheromatous plaque. Consequently, OxPLs induce inflammatory responses by increasing secretion of inflammatory cytokines and chemokines, and through mediation of monocyte activation [21]. Removal of apo(a) from Lp(a) particles leads to increased Lp-PLA_2_ activity, and thus degradation of OxPLs [46].

Lipoprotein(a) also carries monocyte chemoattractant protein-1 (MCP-1), and it has been shown that OxPLs are major determinants of MCP-1 binding. It was suggested that Lp(a)-associated MCP-1 enhances recruitment of monocytes to the vascular wall [47]. Monocyte activation has a pivotal role in atheroma plaque formation, while apo(a) enhances the inflammatory and proteolytic potential of monocytes, which release reactive oxygen species and matrix metalloproteinase-9 (MMP-9). Reactive oxygen species are involved in LDL cholesterol oxidation and formation of foam cells, and MMP-9 contributes to extracellular matrix degradation and rupture of atherosclerotic plaques. The degree of stimulation with apo(a) is inversly correlated with the number of KIV_2_ repeats, which supports the hypothesis regarding the detrimental effects of small-sized isoforms of apo(a) in atherosclerosis progression [48].

According to Arai et al. [49], OxPLs are mainly bound to small Lp(a) isoforms. Carriers of the rs3798220 variation have significantly higher Lp(a) levels and smaller apo(a) isoforms, compared to noncarriers. Patients with this variation are associated with greater levels of proinflammatory OxPLs on apoB particles, and consequently, they might have increased atherogenic potential [49].

A secondary post-hoc analysis of a double-blind, randomized clinical trial (‘Assesment of Clinical Effects of Cholesteryl Ester Transfer Protein Inhibition With Evacetrapib in Patients at a High Risk for Vascular Outcomes’; ACCELERATE) showed that increased Lp(a) levels are related to cardiovascular events when high sensitivity C-reactive protein (CRP) concentrations are ≥2 mg/L, but not when they are <2 mg/L [50]. The inflammatory marker CRP directly affects endothelial cells, monocytes, macrophages, and smooth muscle cells. It is produced by hepatocytes in response to various inflammatory cytokines. Among these, IL-6 is most closely associated with inflammation and increased risk of atherosclerosis. As shown by Shufta et al., there is a positive correlation between CRP, IL-6, and Lp(a) levels [51]. Furthermore, the IL-6 receptor blocker tocilizumab reduced inflammatory disease activity and plasma Lp(a) levels in 132 patients with reumatoid arthritis [52]. Also, in a study of 280 hemodialysis patients and in two studies with 137 and 114 patients with rheumatoid arthritis, CRP levels increased with increasing levels of Lp(a) [53,54,55]. However, based on the ‘Copenhagen General Population Study’, Langsted et al. reported that elevated Lp(a) levels are not causally associated with increased low-grade inflammation, as measured through CRP, despite a casual association with increased risk of aortic valve stenosis and acute myocardial infarction [56].

## 4. Lipoprotein(a) and Atherothrombosis

Coagulation and fibrinolysis have essential roles in atherothrombosis [57]. The rupture of an atherosclerotic plaque results in the activation of platelets and the extrinsic coagulation pathway. Lp(a) participates in atherothrombosis through several mechanisms (Figure 3). As an atherogenic lipoprotein, Lp(a) interferes with platelet aggregation [58], as it can bind to platelet-activating factor (PAF) acetylhydrolase, which degrades and inactivates PAF. This mechanism results in reduced platelet aggregation and activation [59].

Lipoprotein(a) has an effect on the coagulation pathway through the promotion of the expression of tissue factor (TF) [60]. TF is a cell-surface glycoprotein that is overexpressed by macrophages and smooth muscle cells within the atherosclerotic plaque. Tissue damage that results in endothelial denudation exposes TF to the flow of blood, whereby TF initiates activation of the extrinsic coagulation pathway, which leads to thrombus formation and intima fibrin deposition [57]. Another prothrombotic effect of Lp(a) is related to its binding to and inhibition of the activity of the TF pathway inhibitor (TFPI) [61]. Bilgen et al. and Nisio et al. showed positive correlations between plasma TFPI and Lp(a) levels [62,63]. Although patients with higher Lp(a) levels had higher TFPI concentrations, the TFPI activity was not different between those with high and low Lp(a) levels. This suggests that the binding of TFPI to Lp(a) partly inhibits TFPI activity [62].

Lipoprotein(a) is believed to promote atherothrombosis due to its homology with plasminogen [5]. As indicated, one of the component proteins of Lp(a), apo(a), shares more than 80% of its protein sequence with plasminogen [64]. Lp(a) lacks KI to KIII of plasminogen, but it contains KIV and KV [60]. Apo(a) KIV_10_ contains a lysin binding site, which is most similar to the lysine binding site within plasminogen KIV [65]. Due to this structural homology, Lp(a) can bind to plasminogen receptors on the surface of platelets and prevent the interaction between plasminogen and tissue plasminogen activator (tPA). Therefore, tPA cannot convert plasminogen to plasmin [60].

Plasminogen is the proenzyme precursor of plasmin, and thus it has an important role in fibrinolysis. Plasminogen is activated to plasmin by tPA or urokinase [66]. The plasminogen and *LPA* genes are both on chromosome 6, and a recent study identified nine SNPs within the plasminogen and *LPA* gene region that are significantly associated with plasminogen levels [67]. Among these various SNPs, Wang et al. studied the impact of rs3798220 and rs10455872 on Lp(a) and plasminogen levels. Although these polymorphisms are associated with higher Lp(a) levels, they had no influence on plasminogen levels and fibrinolytic activity [68]. However, in an in vitro study, Rowland et al. showed that Caucasian patients carrying the rs3798220 SNP have increased clot lysis times and reduced clot permeability, compared to noncarriers. On the other hand, among non-Caucasians, increased clot permeability and decreased lysis time have been reported [69]. Furthermore, Scipione et al. showed that the isoleucine-to-methionine substitution in apo(a) protease-like domain of rs3798220 decreased coagulation times and increased fibrin clot lysis times, compared to wild-type apo(a). Additionally, the fibrin fiber width in plasma clots was increased, and patients with higher Lp(a) levels had lower clot permeability [70]. Taken together, rs3798220 has antifibrinolytic potential and is associated with increased Lp(a) levels and cardiovascular risk.

Lipoprotein(a) inhibits the production of tPA, and therefore it promotes a thrombotic state through the prevention of plasmin-mediated clot lysis [60]. In addition, Lp(a) stimulates the expression and activity of the primary inhibitor of the fibrinolytic system, plasminogen activator inhibitor-1 (PAI-1) [60,71]. We have previously shown that patients with familial hyperholesterolemia who had survived myocardial infarction at a young age had significanly decreased fibrinolytic activity, as shown by increased PAI-1 antigen and activity, and increased t-PA antigen, compared to those without myocardial infarction [72]. Furthermore, these patients had higher Lp(a) levels; however, this difference did not reach significance, probably due to the small number of patients. Etingin et al. performed an in vitro study in which they treated human umbilical vein endothelial cells with Lp(a) and LDL. They showed that Lp(a) levels up to 40 µg/mL increased PAI-1 activity, while treatment with LDL cholesterol using approximately equimolar concentrations did not stimulate PAI-1 activity. Lp(a) levels above 40 µg/mL did not show any further increase in PAI-1 activity. This effect was due to increased mRNA levels of PAI-1 [73]. Shindo et al. compared coronary atherotomy specimens of patients with acute myocardial infarction and unstable angina pectoris with those from patients with stable angina pectoris, and showed that the levels of both Lp(a) and PAI-1 were significantly higher for the former [74]. PAI-1 activity, but not antigen, was shown to be a predictive factor for future coronary events in patients with or without prior CAD [75]. Some studies have also included Lp(a) and apoB in the analysis of fibrinolytic parameters, and in all of these, the fibrinolytic parameters were correlated with Lp(a) or apoB, which is a component of Lp(a) (Table 1) [76,77,78].

Fibrin represents the structural framework for fibrin clots. The structure, stability, and fibrinolytic potential of fibrin clots depend on differences in the fibrin fiber diameter and pore size within the fibrin network, and on negatively charged substances. Dense clots are associated with a reduced fibrin fiber diameter, smaller pores, and increased fibrin stiffness [79]. The rate of fibrin clot lysis is predominantly a function of pore size. Fibrin clots with smaller pores increase clot lysis times and are correlated with greater cardiovascular risk. Elevated Lp(a) levels are correlated with reduced fibrin clot permeability and impaired fibrinolysis [61]. Lp(a) competes with plasminogen and tPA for binding sites on fibrin, and consequently, it impairs fibrinolysis [1,60,64]. These antifibrinolytic effects are more profound for smaller-sized isoforms of Lp(a), because these have a higher affinity for their binding to fibrin [80]. The predictive value of high Lp(a) levels as a risk factor therefore depends on the concentration of Lp(a) particles containing small isoforms of Lp(a) [81]. Other reports have opposed the mechanism whereby Lp(a) directly competes with plasminogen for binding to fibrin, and have indicated that apo(a) forms a quaternary complex with plasminogen, tPA, and fibrin [82,83]. In addition, the negatively charged apo(a) might affect the Lp(a)-induced alteration of fibrin clot structure, and thus contribute to reduced fibrin degradation [61].

## 5. Prognostic Value of Lipoprotein(a) in Combination with Thrombotic and Inflammation Parameters

The prospective studies that have included analysis of Lp(a) and at least one marker of inflammation and/or coagulation/fibrinolysis as predictors for future coronary events are presented in Table 1. In primary prevention studies, which have included >40,000 patients (mostly male), Lp(a) was shown to independently predict the first cardiovascular event [84,85,86]. In the first study, by Langsted et al. [84], only Lp(a) and CRP were examined as markers of inflammation. In the next two studies, by Cremer et al. [85] and Seed et al. [86], only Lp(a) and fibrinogen were examined as markers of impairment of the coagulation fibrinolytic system. Indeed, none of the studies indicated in Table 1 have examined Lp(a) plus at least one parameter of inflammation and one of coagulation/fibrinolysis.

For patients with stable coronary disease, their D-dimer and fibrinogen levels increased with the quartiles of CRP, while no such relation was seen for Lp(a) [87]. In a survey of 1172 apparently healthy men, CRP levels correlated with several fibrinolytic parameters (i.e., fibrinogen, t-PA antigen, D-dimer, homocysteine), and also with Lp(a) [88]. Here, we can conclude that fibrinolytic and inflammatory parameters are interconnected in patients with stable CAD and in apparently healthy men.

As the rupture of a pre-existing atherosclerotic plaque is responsible for the majority of acute coronary events, the number of such events in secondary prevention studies is much higher. In these studies, where no coronary interventions were performed due to stenosis < 50%, Lp(a) alone, and even more so in combination with fibrinogen, was an independent predictor of major adverse cardiovascular events after 37 months [89]. This was expected, because in these patients, both progression of atherosclerosis and rupture of plaques can lead to future cardiovascular events. In patients with only percutaneous transluminal coronary angioplasty and no stent implantation due to CAD, t-PA was shown to be a superior predictor of major adverse cardiovascular events after 13 years, compared to Lp(a) [90]. This is not surprising, as the endothelium in percutaneous transluminal coronary angioplasty with no stent implantation is highly disrupted, and future coronary events are mostly due to intracoronary thrombosis. Similarly, in patients six months after percutaneous transluminal coronary angioplasty, neither Lp(a) nor PAI-1 were predictors of restenosis [91]. The reason here might be the short time of observation. In studies where patients were mostly treated with drug-eluting stents, Lp(a) and CRP were independent predictors of future cardiovascular events [92,93]. At first sight, this might be a surprise, but it can be noted that all these patients had dual antiplatelet therapy that prevented early in-stent thrombosis, for which coagulation and fibrinolysis are responsible. Furthermore, sirolimus- and everolimus-eluting stents are less prone to early in-stent thrombosis. In contrast, for patients treated with bare metal stents, homocysteine levels were linked to the fibrinolytic parameters, especially with t-PA and PAI-1 [94,95]. These patients were also treated with dual antiplatelet therapy, although bare metal stents were much more thrombogenic than drug-eluting stents [96]. On the other hand, there were no effects on cardiovascular mortality [97]. Finally, plasma levels of t-PA and Lp(a) were independently associated with the subsequent development of a first myocardial infarction in a prospective case-control study [98].

## 6. Conclusions

Lipoprotein(a) has been shown to be an independent predictor for future cardiovascular events in primary and secondary prevention. This has been confirmed in epidemiological and Mendelian randomized trials, and in the last two years, also in randomized placebo-controlled double-blind trials with PCSK9 inhibitors. Although these drugs were primarly designed to target LDL cholesterol, they also decrease Lp(a) by 20% to 40% [100,101]. Similar findings were observed with inclisiran, a small interfering RNA (siRNA). Inclisiran was developed to target LDL and apoB, but in the phase 2 study entitled Trial to Evaluate the Effect of ALN–PCSSC Treatment on Low Density Lipoprotein Cholesterol (ORION-1), it also reduced Lp(a) levels by 25.6% [102,103]. Oplasiran, another siRNA targeting Lp(a), significantly reduced Lp(a) levels in the phase 1 trial [104]. Moreover, these effects persisted for more than six months [104]. Furthermore, promising Lp(a)-lowering effects have been observed with antisense oligonucleotide AKCEA–APO(a)–LRX. This drug has shown a dose-dependent reduction in Lp(a) levels up to 80% among patients with established cardiovascular disease and Lp(a) levels above 60 mg/dL in a phase 2 trial [105]. Whether viral-mediated gene therapy RGX–501 for homozygous familial hypercholesterolemia also has influence on reducing Lp(a) levels is currently being investigated in clinical trial NCT02651675 [106]. Although all the aforementioned therapeutics have shown lipid-related effects, only PCSK9 inhibitors were found to be associated with decreased cardiovascular events [31,32]. As for their effects on inflammation and coagulation, the current evidence is scarce.

The atherogenic propensity of Lp(a) arises as a consequence of its structure. Its LDL-like particle, which is the main part of Lp(a), has similar effects to those of LDL cholesterol, and it is even more atherogenic because of its higher sensitivity to oxidation. On the other hand, apo(a) has a very similar structure to plasminogen, and because of this, it interferes with the coagulation and fibrinolytic system. Apo(a) thus increases coagulation and decreases fibrinolysis. The third component of apolipoprotein(a), the OxPLs, is responsible for its proinflammatory effects.

In the majority of prospective studies that have included patients after acute coronary syndrome who have undergone bare metal or drug-eluting stent implantation or percutaneous transluminal coronary angioplasty with no stent implantation, the fibrinolytic parameters appear to be more powerful predictors of future cardiovascular events. On the other hand, in patients with no clinically known cardiovascular disease, Lp(a) was a better predictor of future cardiovascular events. These appear to make sense, as in the first group of patients, they already had atherosclerotic plaques prone to rupture, while in the second group of patients, the formation of such plaques was not complete. It is known that for the rupture of vulnerable plaques, disruption of the coagulation fibrinolytic equlibrium is responsible for thrombus formation and a consequent acute coronary event. The formation of such plaques is not only dependent on Lp(a) levels; rather, it is more likely dependent on the presence of higher Lp(a) levels regardless of the LDL cholesterol levels.

The question of whether only a decrease in Lp(a) results in decreased cardiovascular events will be answered by the study ‘Assessing the Impact of Lipoprotein(a) Lowering with TQJ230 on Major Cardiovascular Events in Patients with Cardiovascular Disease’ (HORIZON; NCT04023552), which will end in 2024. However, to answer which of these three pathophysiological mechanisms are affected by Lp(a) lowering, further studies will be needed.

## Figures and Tables

**Figure 1 biomolecules-12-00026-f001:**
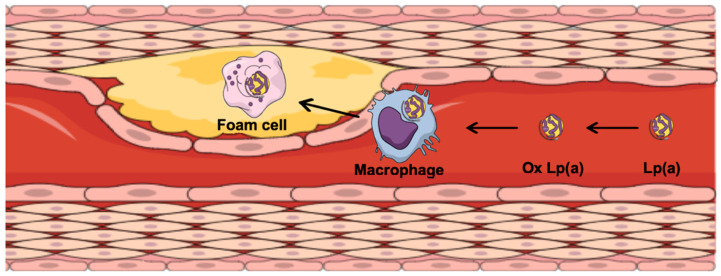
The role of lipoprotein(a) (Lp(a)) in plaque formation. Ox Lp(a), oxidized Lp(a).

**Figure 2 biomolecules-12-00026-f002:**
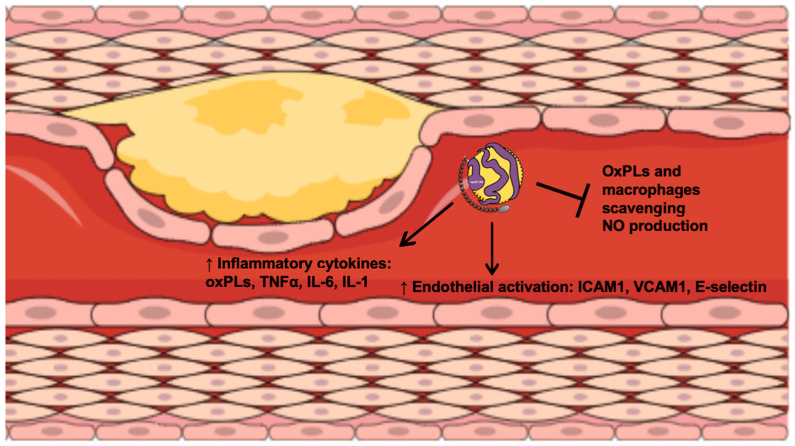
The role of Lp(a) in inflammation. OxPLs, oxidized phospholipids; TNFα, tumor necrosis factor-α; IL, interleukin; ICAM1, intercellular adhesion molecule-1; VCAM1, vascular cell adhesion protein-1; NO, nitric oxide.

**Figure 3 biomolecules-12-00026-f003:**
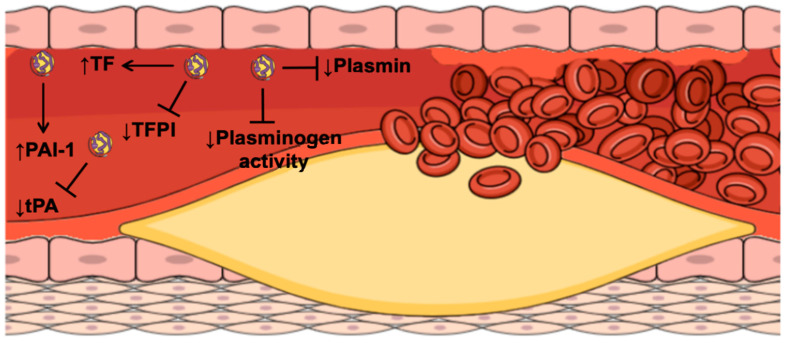
The role of Lp(a) in thrombus formation following plaque rupture. TF, tissue factor; TFPI, tissue factor pathway inhibitor; PAI-1, plasminogen activator inhibitor-1; tPA, tissue plasminogen activator.

**Table 1 biomolecules-12-00026-t001:** Overview of prospective studies that have examined Lp(a) and at least one inflammatory and/or fibrinolytic parameter.

Study	Parameter Included	Study Population (n)			Primary Endpoint	Independent Predictor	Ref.
		(n)	Age (Years)	Characteristics			
Langsted et al. (2014)	Lp(a), CRP	34,829	-	General population	Ischemic heart disease	Lp(a), independent of CRP levels	[84]
Cremer et al. (1997)	Lp(a), fibrinogen	5790	40–60	Male, without previous CAD	MI after 10 years	Lp(a)	[85]
Seed et al. (2001)	Lp(a), fibrinogen	2616	51–61	Male, without previous CAD	CAD after 6 years	Lp(a)	[86]
Zhang et al. (2020)	Lp(a), fibrinogen	8417	-	Stable CAD, no stent implantation	MACE after 37 months	Lp(a), fibrinogen; combination of both superior	[89]
Niessner et al. (2003)	Lp(a), tPA	141	-	CAD after PTCA, no stent implantation	MACE after 13 years	t-PA	[90]
Alaigh et al. (1998)	Lp(a), PAI-1	163	-	CAD after PTCA, no stent implantation	Restenosis after 6 months	None	[91]
Kardys et al. (2012)	Lp(a), CRP, IL-10	161	-	CAD after sirolimus-eluting stent implantation	MACE after 1, 6 years	1 year: Lp(a); 6 years: CRP	[92]
Zairis et al. (2002)	Lp(a), CRP	483	-	ACS after PCI, various stent implantations	MACE after 3 years	Lp(a), CRP	[93]
Marcucci et al. (2006)	Lp(a) PAI-1, homocysteine	520	-	ACS, bar metal stent implantation	MACE after 24 months	PAI-1, homocysteine	[95]
Thogersen et al. (2003)	Lp(a), PAI-1, t-PA, leptin	62, plus 124 controls	Sex and age matched	-	First myocardial infarction	Lp(a), t-PA	[98]
Pineda et al. (2010)	Lp(a), CRP, PAI-1, t-PA, fibrinogen, D-dimer, homocysteine	142; 56; 10	-	MI before 45 years; treated with fibrinolysis; PCI	MACE after 36 months	Homocysteine	[99]
Moss et al. (1999)	Lp(a), PAI-1, fibrinogen, D-dimer	1045	-	Post-MI (2 months), treated with thrombolysis or PCI	Coronary death or nonfatal MI after 26 months	D-dimer	[76]

ACS, acute coronary syndrome; CAD, coronary artery disease; CRP, C-reactive protein; MACE, major adverse coronary event; MI, myocardial infarction; PCI, percutaneous coronary intervention; PTCA, percutaneous transluminal coronary angioplasty.

## Data Availability

Not applicable.

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
