# Peer review of "Lipoprotein(a)—The Crossroads of Atherosclerosis, Atherothrombosis and Inflammation"

_biomolecules, 2021, doi:10.3390/biom12010026_

Round 1
Reviewer 1 Report
This is an the very elegant and comprehensive review of the Lipoprotein(a) and the crosslink between atherosclerosis, atherothrombosis and inflammation. The Authors focused on the available basic science publication and clinical trials presenting us the current knowledge in this field . Reading this paper we will get a very clear message about genetics and physiological role of lipoprotein(a) in different clinical circumstances. Especially very well discussed and documented is part focused on prognostic value of lipoprotein(a) in combination with thrombotic and inflammation parameters. The Authors convinced us that Lipoprotein(a) has been shown to be an independent predictor for future cardiovascular events in primary and secondary prevention Additionally accordingly to genetic and epidemiological studies, Lp(a) is considered pro-atherosclerotic, pro-inflammatory, pro-thrombotic, anti-fibrinolytic, All this information based on profound literature overview we will find in this well written review.
Author Response
Reviewer #1
This is an the very elegant and comprehensive review of the Lipoprotein(a) and the crosslink between atherosclerosis, atherothrombosis and inflammation. The Authors focused on the available basic science publication and clinical trials presenting us the current knowledge in this field. Reading this paper we will get a very clear message about genetics and physiological role of lipoprotein(a) in different clinical circumstances. Especially very well discussed and documented is part focused on prognostic value of lipoprotein(a) in combination with thrombotic and inflammation parameters. The Authors convinced us that Lipoprotein(a) has been shown to be an independent predictor for future cardiovascular events in primary and secondary prevention Additionally accordingly to genetic and epidemiological studies, Lp(a) is considered pro-atherosclerotic, pro-inflammatory, pro-thrombotic, anti-fibrinolytic, All this information based on profound literature overview we will find in this well written review.
Authors’ response: We would like to thank the Reviewer for the time and effort put in here to revise our manuscript.
Reviewer 2 Report
The review about Lp(a) from Ugovsek et al is a clear and concise review about the current progress in Lp(a) research giving enough background for the reader to understand the topic. I would propose to add an outlook specifically for treatment options including options currently under clinical trial (eg RGX-501) or results and benefits/drawbacks of reducing Lp(a) using PCSK9 inhibitors or ezetimibe.
Author Response
Reviewer #2: The review about Lp(a) from Ugovsek et al is a clear and concise review about the current progress in Lp(a) research giving enough background for the reader to understand the topic. I would propose to add an outlook specifically for treatment options including options currently under clinical trial (eg RGX-501) or results and benefits/drawbacks of reducing Lp(a) using PCSK9 inhibitors or ezetimibe.
Authors’ response:
We would like to thank the Reviewer for the time and effort put in here to revise our manuscript. Thank you also for your valuable comment. We have carefully addressed your concern (please see our response below), and revised our manuscript accordingly. All of the changes made to the original manuscript are visible as tracked changes in the revised manuscript.
Based on the Reviewer’s suggestion to add an outlook for promising future treatment options for lowering Lp(a), we have added the following paragraph under Conclusions: “Similar findings were observed with inclisiran, a small interfering RNA (siRNA). Inclisiran was developed to target LDL and apoB, but in the phase 2 study entitled Trial to Evaluate the Effect of ALN-PCSSC Treatment on Low Density Lipoprotein Cholesterol (ORION-1), it also reduced Lp(a) levels by 25.6% [102], [103]. Oplasiran, another siRNA targeting Lp(a), significantly reduced Lp(a) levels in the phase 1 trial [104]. Moreover, these effects persisted for more than 6 months [104]. Furthermore, a promising Lp(a)-lowering effects have been observed with antisense oligonucleotide AKCEA-APO(a)-LRX. This drug has shown a dose-dependent reduction in Lp(a) levels up to 80% among patients with established cardiovascular disease and Lp(a) levels above 60 mg/dL in a phase 2 trial [105]. Whether viral-mediated gene therapy RGX-501 for homozygous familial hypercholesterolemia also has influence on reducing Lp(a) levels, is currently being investigated in clinical trial NCT02651675 [106]. Although all the aforementioned therapeutics have shown lipid-related effects, only PCSK9 inhibitors were found to be associated with decreased cardiovascular events [31] [32]. As for their effects on inflammation and coagulation, the current evidence is scarce.”
In our opinion ezetimibe does not fit within the scope of this review as it lacks the effects on Lp(a). Therefore we have only mentioned PCSK9 inhibitors.